# Assessing the Prognostic Value of Cytoplasmic and Stromal Caveolin-1 in Early Triple-Negative Breast Cancer Undergoing Neoadjuvant Chemotherapy

**DOI:** 10.3390/ijms252212241

**Published:** 2024-11-14

**Authors:** Iris Teruel, Eva Castellà, Sabela Recalde, Gemma Viñas, Anna Petit, Macedonia Trigueros, Eva Martínez-Balibrea, Eudald Felip, Milana Bergamino, Adrià Bernat-Peguera, Beatriz Cirauqui, Vanesa Quiroga, Angelica Ferrando-Díez, Anna Pous, Assumpció López, Laia Boronat, Gemma Soler, Jordi Recuero, Margarita Romeo, Pau Guillén, Ricard Mesía, Ester Ballana, Anna Martínez-Cardús, Mireia Margelí

**Affiliations:** 1Medical Oncology Department, Catalan Institut of Oncology (ICO)-Badalona, B-ARGO (Badalona Applied Research Group in Oncology) and IGTP (Health Research Institute Germans Trias i Pujol), Universitat Autònoma de Barcelona, 08916 Badalona, Spain; iteruel@iconcologia.net (I.T.); efelip@iconcologia.net (E.F.); mabergamino@iconcologia.net (M.B.); bcirauqui@iconcologia.net (B.C.); vquiroga@csdm.cat (V.Q.); aferrandod@iconcologia.net (A.F.-D.); apousb@iconcologia.net (A.P.); alopezpa@iconcologia.net (A.L.); lboronat@iconcologia.net (L.B.); gsoler@iconcologia.net (G.S.); jrecuero@iconcologia.net (J.R.); pguillens@iconcologia.net (P.G.); rmesia@iconcologia.net (R.M.); 2Department of Pathology, Hospital Germans Trias i Pujol, IGTP (Health Research Institute Germans Trias i Pujol), Universitat Autònoma de Barcelona, 08916 Badalona, Spain; ecastella.germanstrias@gencat.cat; 3Department of Medical Oncology-Breast Cancer Unit, Institut Català d’Oncologia (ICO)-H.U.Bellvitge, Institut d’Investigació Biomèdica de Bellvitge (IDIBELL), Universitat de Barcelona, 08907 Barcelona, Spain; srecalde@iconcologia.net; 4Department of Medical Oncology-Breast Cancer Unit, Institut Català d’Oncologia (ICO)-H.U.Doctor Josep Trueta, Precision Oncology Group (OncoGIR-Pro), Institut d’Investigació Biomèdica de Girona (IDIBGI), Universitat de Girona, 17007 Girona, Spain; gvinyes@iconcologia.net; 5Departament of Pathology, Hospital Universitari de Bellvitge, Institut d’Investigació Biomèdica de Bellvitge (IDIBELL), L’Hospitalet del Llobregat, 08908 Barcelona, Spain; apetit@bellvitgehospital.cat; 6AIDS Research Institute-IrsiCaixa, Health Research Institute Germans Trias i Pujol (IGTP), Can Ruti Campus, Universitat Autònoma de Barcelona, 08916 Badalona, Spaineballana@irsicaixa.es (E.B.); 7ProCURE Program, Institut Català d’Oncologia (ICO) and CARE Program, Health Research Institute Germans Trias i Pujol (IGTP), 08916 Badalona, Spain; embalibrea@iconcologia.net; 8CARE Program, Catalan Institut of Oncology (ICO)-Badalona, B-ARGO (Badalona Applied Research Group in Oncology) and IGTP (Health Research Institute Germans Trias i Pujol), 08916 Badalona, Spain; abernat@igtp.cat

**Keywords:** CAV1, TNBC, breast cancer prognosis, TME

## Abstract

Triple-negative breast cancer (TNBC) is a highly aggressive subtype with limited therapeutic options, leading to higher relapse rates and mortality. Identifying prognostic biomarkers like caveolin-1 (CAV1) is crucial for personalized treatment. CAV1 influences tumor progression and chemotherapy response, particularly through its interaction with the tumor microenvironment (TME) and cancer metabolism. Understanding the prognostic value of CAV1 in different cellular compartments is essential for its clinical application in TNBC. In the methods section CAV1 gene expression in TNBC was evaluated using in silico analysis, followed by the immunohistochemical staining of tumor cytoplasm (cCAV1) and stromal cells (sCAV1) in 58 early-stage TNBC patients. Statistical analyses were performed to correlate CAV1 expression with clinicopathological features and survival. In the results section, in silico analysis revealed higher CAV1 expression in TNBC, correlating with shorter overall survival. In the patient samples, cCAV1 was observed in 10.3% of cases, and was associated with larger tumors, higher grades, and poorer prognoses. sCAV1 was detected in 42% of cases, associated with less proliferative and less aggressive tumors, but did not significantly impact prognoses. In conclusion, cCAV1 expression is a significant prognostic marker in early-stage TNBC, highlighting the importance of assessing CAV1 in different cellular compartments. Further research is needed to explore the mechanisms and clinical implications of cCAV1.

## 1. Introduction

Triple-negative breast cancer (TNBC) constitutes around 10%-15% of breast cancer (BC) cases and lacks the expression of estrogen receptors, progesterone receptors, and HER2 protein amplification [1]. TNBC, including early-stage cases, poses significant challenges in management due to its aggressive nature and the limited effective suitable therapies, leading to higher relapse rates and increased mortality risks compared to other breast cancer subtypes. Neoadjuvant (NA) chemoimmunotherapy is now widely preferred for stage II or III TNBC treatment, as recommended by the ASCO and St. Gallen guidelines [2,3], achieving a pathologic complete response (pCR) in around 64% of cases [4]. Classically, anthracycline and taxane-based chemotherapy stand as mainstays of TNBC treatment [5]; adding carboplatin boosts pCR rates, particularly in specific patient groups [6]. Recently, the addition of pembrolizumab to chemotherapy has become a new standard for high-risk TNBC, improving pCR rates and event-free survival [7], these being crucial for improving long-term outcomes. Therefore, this NA strategy enhances TNBC beneficial outcomes, also allowing the identification of those patients who do not achieve pCR and would need tailored adjuvant therapies, such as adjuvant capecitabine [8] and platinum agents [9], as well as PARP inhibitors in high-risk TNBC with BRCA mutations [10]. In this scenario, there is a clear need to find prognostic biomarkers that can identify those patients with more aggressive disease in order to design a more effective treatment strategy.

Caveolin-1 (CAV1) is the main protein located in cholesterol-rich plasma membrane raft domains, the caveolae: flask-shaped invaginations of the plasma membrane that are located at the cell surface in most cell types [11]. It participates in different intracellular signaling pathways associated with tumorigenic processes, such as neoplastic transformation, cell survival, proliferation, angiogenesis, and metastasis [12], but also modulates many cellular functions, including nutrient and drug internalization, tumor–stroma interactions, hypoxia response, inflammation, epithelial–mesenchymal transition, and cell cycle regulation. Indeed, its role in tumorigenesis has been studied in different cancer subtypes, showing opposite behaviors to tumor suppressor genes or oncogenes [13]. Although there is convincing evidence from in vitro models supporting the relevance of CAV1 in BC, its actual significance in clinical settings is not yet well established [14,15,16]. The interplay between CAV1 and the tumor microenvironment (TME) is crucial for understanding cancer metabolism. In this context, the loss of stromal CAV1 promotes the “reverse Warburg effect,” having been linked to adverse clinical outcomes [17].

It is noteworthy that the prognostic value of CAV1 expression within the tumor epithelium and stroma remains a subject of contention. Cytoplasmic CAV1 (cCAV1) in tumor cells has been associated with a more malignant phenotype, including correlations with estrogen receptor (ER) negativity, as well as basal-like and BRCAness phenotypes [18,19,20]. The prognostic significance of stromal CAV1 (sCAV1) in BC remains elusive. This ambiguity underscores the complex interplay between CAV1 expression, tumor characteristics, and the evolving TME. Efforts aimed at improving treatment responses in TNBC include the identification of mechanisms influenced by CAV1, with the goal of strengthening the immune response against TNBC cells. This encompasses strategies targeting antigenic mutations, immune modulation, and TME modification to rectify metabolic imbalances associated with CAV1 loss.

Considering the crucial and controversial role played by CAV1 in TNBC progression and treatment, this work attempts to delve into the impact of CAV1 expression in prognosis. Here, we consider different cellular compartments in the tumor and surrounding microenvironment within a homogenous cohort of early-stage TNBC cases, and determine its associations with clinical outcomes and classical clinicopathologic variables. The elucidation of the prognostic value of CAV1 according to its distribution in the different cellular compartments could play a key role in the clinical prognosis of early TNBC. Importantly, the intricate interplay between CAV1, the TME, and cancer metabolism underscores the complexity of cancer biology and opens avenues for novel therapeutic strategies targeting metabolic dependencies in cancer.

## 2. Results

### 2.1. In Silico Analysis of the Impact of CAV1 Expression in TNBC

We used CCLE to explore the mRNA expression data from 57 BC cell lines, classified by molecular subtypes. We found statistically significant higher levels of CAV1 in TNBC cell lines compared to the rest of the subtypes (Student’s T test; *p*-value < 0.05) (Appendix A). Secondly, we evaluated mRNA expression data from 659 patients (406 luminal A, 111 luminal B, 108 TN and 34 HER2+) from the TCGA database using the Wanderer tool, although we did not observe statistically significant differences in CAV1 expression among subtypes (Appendix A). To finish, Kaplan–Meier curves and log-rank tests were performed to compare survival rates in non-metastatic BC patients according to high or low levels of CAV1. We found statistically significant differences between groups, with a shorter OS in the high-expression cohort (HR 2.34, 95%CI 1.18–4.66, *p* = 0.012) (Appendix A). When we selected only TNBC cases, the trend was similar, though statistical significance was not reached (HR 3.78, 95%CI 0.44–32.47, *p* = 0.19). According to our exploratory study, we assumed the potential impact of CAV1 expression in BC aggressiveness and prognosis, focusing on TNBC.

### 2.2. Patients and Tumor Characteristics

The clinicopathologic characteristics seen in our early-stage TNBC cohort of study (N = 58) are listed in Table 1. Briefly, the median age of patients at diagnosis was 52.4 years; 82.8% (48 of 58) of tumors were grade III; median Ki67 percentage expression was 68.7%; 23 patients had positive nodes; and stage II was the most frequent staging at diagnosis. The percentage of tumor-infiltrating lymphocytes (TILs) was detectable in 17 out of the 58 samples, with a median of 24.12%.

All of them received nab-paclitaxel based NA chemotherapy before surgery: there were 26 cases in monotherapy and 46 followed by sequential anthracyclines plus cyclophosphamide treatment. Adjuvant radiotherapy was indicated in 49 patients.

In terms of outcomes, pCR was achieved in 20 out of 58 (34.5%) patients. With a median follow-up time of 51.43 months (min–max 9–74), in accordance with the median follow-up reported in trials with a similar population [7], 15 recurrences were reported (median time to relapse 22.06 months, 95%CI 16.27–27.86). Median RFS and OS were 60.22 (95%CI 54.1–66.34) and 62.01 months (95%CI 56.57–67.44), respectively.

### 2.3. Immunohistochemical Evaluation of CAV1 Protein Expression Stratified by Cellular Compartments

The evaluation of CAV1 protein expression was performed by immunohistochemistry according to different cellular compartments: cytoplasmatic expression in tumor cells or cCAV1, and expression in stromal cells or sCAV1. The positive expression of cCAV1 (+cCAV1) was detected in 10.3% (6 out of 58) of cases; sCAV1 evaluation was successfully performed in 57 of the 58 samples, finding positive expression in 24 of them (+sCAV1) (42%). For combined CAV1 expression score grading, 5.2% (3 out of 57) of cases were both cCAV1- and sCAV1-positive, while 47.37% (27 out of 57) were positive for either of them and 52.63% (30 of 57) were negative.

### 2.4. Association Between Levels of CAV1 and Patient’s Characteristics

The association of CAV1 compartmental protein expression and patients’ covariables are described in Appendix A. Summarizing, the +cCAV1 group was associated with more aggressive cases and a trend to bigger tumors, all of them grade III and more frequently presenting vascular invasion in the residual tumor after NA chemotherapy (*p* = 0.01). On the other hand, the +sCAV1 group showed an enrichment of tumors with lower levels of Ki67 (*p* = 0.07) and fewer TILs (*p* = 0.06). Considering BRCA mutation status, we found 9 patients with a pathogenic or likely pathogenic variant in the *BRCA1* gene. Interestingly, none of those cases expressed cCAV1. In contrast, we found +sCAV1 in 5 out of the 9 cases (65.5%), which did not reach the level of statistical significance. In a global analysis of CAV1 expression, considering positive those cases which were positive for at least one or both cCAV1 and sCAV1, neither demonstrated a statistically significant association with the clinical characteristics of patients.

### 2.5. Impact of Levels of CAV1 in Response to Treatment and Prognosis

In our early-stage TNBC cohort, none of the 20 patients who experienced pCR to NA treatment were +cCAV1. Four +cCAV1 cases relapsed during the follow-up period (median 51.43 months) (Fisher test, *p* = 0.034), establishing an association between +cCAV1 and poorer prognoses. Similarly, +cCAV1 showed a statistically significant impact in RFS, with the median time in the +cCAV1 group being shorter than in negative ones (31.33 vs. 63.09 months; 95%CI 1.39–32.6, *p* = 0.002). Similar results were observed in OS, which was also poorer in +cCAV1 groups (35.16 vs. 64.65 months; 95%CI 8.59–49.4, *p* = 0.02) (Figure 1).

According to Cox univariate analysis, +cCAV1 expression (p = 0.006), tumors stage III at diagnosis (*p* = 0.001), the presence of affected lymph nodes (p = 0.022), NA with nab-paclitaxel alone (*p* = 0.038), and non-pCR (*p* = 0.038) were shown to be prognostic factors of worse prognosis regarding RFS, and resulted in worse OS (Table 2). Interestingly, in the Cox multivariate analysis, +cCAV1 expression was shown to be an independent prognostic factor for shorter RFS (*p* = 0.049) and OS (*p* = 0.04) (Figure 2). No association between +sCAV1 levels and prognosis was found, including when global CAV1 expression was considered.

## 3. Material and Methods

### 3.1. In Silico Analysis of the Impact of CAV1 Expression in TNBC

To determine the levels of CAV1 expression in BC, we used mRNA expression data from BC cell lines and patients available in the Cancer Cell Line Encyclopedia (CCLE) (https://portals.broadinstitute.org/ccle; accessed 15 November 2018) and the Tumor Cancer Genome Atlas (TCGA), respectively. Both were classified by molecular subtypes and their association with CAV1 levels was assessed by Student’s T test analysis. For survival analyses, patients’ data were analyzed using the Wanderer tool (maplab.imppc.org/wanderer/; accessed 15 November 2018). Kaplan–Meier curves and log-rank tests were performed with the c-bioportal (www.cbioportal.org; accessed 15 November 2018), considering the median as the threshold to establish subgroups of CAV1 expression (high vs. low). A *p*-value under 0.05 was considered statistically significant.

### 3.2. Patients and Samples

We retrospectively collected treatment-naïve FFPE tumor biopsies at diagnosis, from 58 patients with early-stage TNBC from Institut Catala d’Oncologia (ICO) centers (ICO Badalona, ICO Hospitalet and ICO Girona), diagnosed between 2016 and 2018. All patients received NA chemotherapy: this involved nab-paclitaxel (NP) 125 mg/m^2^ followed by anthracycline (epirubicin 90 mg/m^2^) and cyclophosphamide 600 mg/m^2^ (EC), or the use of NP 125 mg/m^2^ in monotherapy. Surgery and additional radiotherapy were performed according to the clinical guidelines of the institution [21]. PCR was defined by the absence of a residual infiltrating tumor in the breast and axilla after NA therapy.

This study was conducted in accordance with the ethics principles of the Declaration of Helsinki and approved by the Research and Ethics Committee of Hospital Germans Trias i Pujol (PI-19-062). Samples were obtained from the Biobank where they had previously been pseudo-anonymized, in accordance with the European Data Protection Regulation (EU) 2016/679. All patients provided written informed consent to participate in the study.

The evaluation of *BRCA* germline mutations was carried out following the clinical criteria set out by the Genetic Counselling Units of ICO centers.

### 3.3. Immunohistochemical Staining of CAV1

Immunohistochemical staining was performed in the entire cohort (N = 58), and, in brief, it was conducted on 4-μm thick sections of FFPE tissue mounted on electrically charged slides, dewaxed, and rehydrated in a Benchmark Ultra automatic Ventana System. Heat-induced antigen retrieval was conducted (52 min) in a microwave oven in CC-1 at high pH values. Then, tissue sections were incubated with the primary antibodies. Monoclonal antibody D46G3 XP Rabbit mAb was used for CAV1, at a dilution 1/800. All immunostaining had an internal positive control for CAV1 (vascular endothelial cells and adipocytes).

### 3.4. Immunohistochemical Analysis

CAV1 protein expression was evaluated in cells from both stromal and tumor compartments by two independent observers. The staining was scored semi-quantitatively as negative (0; no staining) or positive (1; diffuse weak staining or strong staining in more than 10% cells per core) for each category (Figure 3).

### 3.5. Statistical Analysis

For statistical analyses, IBM^®^ SPSS Statistics^®^ version 28.0.2.0 was used. The cohort’s description was analyzed based on demographic, clinical, and biological data. All categorical variables were summarized through frequencies and percentages. For quantitative variables, means and standard errors were used. We applied different statistical functions to compare covariables: Mann–Whitney U or Student’s T tests to compare qualitative vs. quantitative; Chi-Square or Fisher’s tests for qualitative variables; and Spearman’s correlation for quantitative ones. A *p*-value under 0.05 was considered statistically significant.

Regarding the survival analysis, the endpoints analyzed in each case were relapse-free survival (RFS, defined as the time from date of cohort treatment initiation to date of relapse) and overall survival (OS, defined as the time from date of cohort treatment initiation to date of death). At date of last follow-up, patients who were disease-free (in RFS) or alive (in OS) were censored. Analyses were conducted using Kaplan–Meier curves and log-rank tests. We estimated median times for RFS and OS, with their confidence intervals (CI) at the 95% level. Cox proportional hazards models were used for univariate and multivariate survival analyses. A *p*-value under 0.05 was considered statistically significant.

## 4. Discussion

The role of CAV1 in tumor development is intricate and controversial. While *CAV1* has long been considered a tumor suppressor gene, recent studies have proposed a different perspective on its role in tumors. For instance, in breast tumors, CAV1 appears to shield tumor cells from cell death, promotes their survival, and prevents the activation of certain tumor-suppressing mechanisms [22,23]. Our in silico explorative analysis supports this idea, observing a strong association between CAV1 overexpression and poor prognosis in an early-stage BC cohort, using data from the TCGA database. This is also concordant with other studies in the literature, showing CAV1’s mRNA expression to be higher in TNBC cell lines compared with other subtypes. In this vein, a recent publication has also drawn attention to an association between CAV1 and the basal subtype, as well as epithelial–mesenchymal transition and BRCAness phenotypes—characteristics of TNBC—when examining TCGA and GOBO datasets [24].

Nevertheless, the evidence from other available studies would suggest that CAV1 could act as a suppressor in early stages of cancer but become a promoter during metastasis. Indeed, its role could differ depending on the presence or absence of certain proteins, tissue types, and within the TME [25]. Taking the controversial role of CAV1 function in cancer development, as tumor suppressor or oncogene, in the present study, we focused on elucidating its impact in TNBC prognosis according to the intratumoral compartment where it is expressing: in cytoplasm of tumor cells or in tumor stroma. CAV1 expression was evaluated semi-quantitatively in tumor tissue from a homogeneous cohort of early-stage TNBC patients, detecting +cCAV1 and +sCAV1 in 10.3% and 42% of cases, respectively. These results are consistent with those reported previously, in a retrospective cohort of 100 early BC patients, including different subtypes, where a positivity rate of 17% was reported for cCAV1, while 57% showed positivity for sCAV1 [26]. Moreover, the percentages observed when analyzing c/sCAV1 together describe similar proportions, in agreement with other studies [20,27].

Looking at +cCAV1 tumors, its expression showed correlation with clinical characteristics of more tumor aggressiveness and lack of pCR after NA treatment. In this regard, some groups have also found a correlation between +cCAV1 and a more malignant tumor phenotype, including a strong association with a lack of expression of estrogen receptors, epidermal growth factor receptor (EGFR) overexpression and basal-like biomarkers [14,19,24,27,28]. Our findings are also consistent with the worse prognosis of the +cCAV1 group, defining a more aggressive subgroup inside the heterogenicity of TNBC.

Findings similar to ours were reported in a recent population-based study, the BCblood cohort [24], where CAV1 was assessed in samples from 1018 patients with non-metastatic invasive BC, followed up for a median duration of 9.0 years, revealing that +cCAV1 was weakly linked to a higher risk of recurrence. Also, in another cohort, overexpression of cCAV1 in patients with BC treated with NA chemotherapy was associated with shorter disease-free survival and OS [29]. Wrapping up, these findings collectively suggest that cCAV1 expression is a putative prognostic biomarker in BC, establishing itself as an independent factor in the multivariate analyses of both these prior publications and in our own study, focused on TNBC. However, other additional studies with smaller sample sizes have yielded conflicting results [14]. Hence, CAV1’s prognostic impact could be context-dependent, considering not only tumor biology and receptor status, but also the stage of disease presentation (early versus advanced stages) [15,16,30].

We found nine patients with a relevant variant in the *BRCA1* gene in our cohort; none of the tumors had +cCAV1, but five were +sCAV1 (20%). In a previous study, the expression of cCAV1 has been found in 22.2% of BRCA1-related cases, and five of six have been basal-like [19]. It is a fact that BRCA1-related tumors are TN more frequently [31], and previous data show similarities between those and sporadic basal-like breast carcinomas, with regard to morphological and immunohistochemical features [32]. Despite this indirect information, we have not found data to support a significant relationship between the expression of CAV1 and the presence of mutations in the *BRCA1* gene, putatively influenced by the small sample size of the study.

Beyond TNBC, several studies have indicated that high levels of cCAV1 expression are linked to poor clinical outcomes across various solid tumors. For instance, in renal cell carcinoma, elevated CAV1 levels are associated with a high tumor grade, lymph node metastasis, and decreased survival rates [33]. Similar data were observed in prostate cancer, where high CAV1 expression correlates with advanced disease and reduced survival after surgery [34]. Likewise, in esophageal squamous cell carcinoma or pancreatic cancer, CAV1 overexpression is linked to metastasis and worse survival [35,36].

On the other hand, sCAV1 expression showed no significant correlation with any of the clinicopathological parameters, which was consistent with some previous reports [13], but inconsistent with others [37]. In line with this, some authors suggested that high CAV1 expression in tumor cells and a lack of this expression in stromal cells could help identify a particular subgroup of BC patients with potentially poor survival [13]. In our experience, +sCAV1 only showed an association at the borderline of statistical significance with lower median levels of Ki67 and with reduced levels of TILS, in accordance with data published by other groups, in which loss of sCAV1 has been associated with worse prognosis [38] and a basal-like subtype in TNBC patients [39]. Several studies have clearly demonstrated that a loss of sCAV1 in fibroblasts is sufficient to induce a CAF phenotype, a fibroblast’s subtype with a hyperproliferation phenotype that has demonstrated the capability to boost tumor growth and generate highly vascularized tumors in vitro [38]. Conversely, high levels of sCAV1 surrounding the tumor have been associated with reduced metastasis and improved survival (*p* < 0.0001) [40]. These data collectively suggest that CAV1 could function also as a tumor suppressor, depending on the type of cells where it is expressed in the tumor and TME, and probably also depending on the tumor stage [41]. Overall, our results add new evidence to the role of CAV1 in TNBC but have to be taken with caution, due to the limited sample size.

In essence, our study highlights that the prognostic impact of CAV1 in early TNBC patients is highly dependent on the tumor compartment where it expresses: the cytoplasm of tumor cells or tumor stroma. Notably, +cCAV1 is closely tied to worse prognosis in primary TNBC cases, suggesting a role in driving rapid disease progression within this specific subgroup. In this context, studies in cell lines have shown that the overexpression of CAV1 in BC cells is associated with an increase in mitochondrial fission, promoting migration [42]. Although sCAV1 is thought to have a borderline correlation with less proliferative tumors, further research into the molecular mechanisms behind sCAV1 expression, and its interactions within the TME, is necessary. Additionally, targeting cCAV1 expression could be postulated as an emerging and promising therapeutic strategy. However, the impact of sCAV1 expression should be elucidated, as its silencing could be dramatic and counterproductive in a TNBC scenario. It is important to note the association between CAV1 expression and the lack of pCR after administration of NA nab-paclitaxel based chemotherapy. These findings could suggest the need for some alternative therapeutic management in TNBC with +cCAV1, for which achieving a pCR is crucial for improving prognosis and survival.

## Figures and Tables

**Figure 1 ijms-25-12241-f001:**
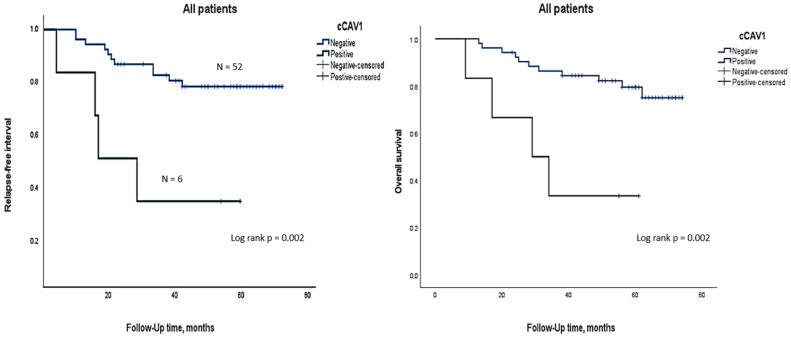
Kaplan–Meier plots of relapse-free survival (RFS), left, and overall survival (OS), right, analysis of cCAV1 expression.

**Figure 2 ijms-25-12241-f002:**
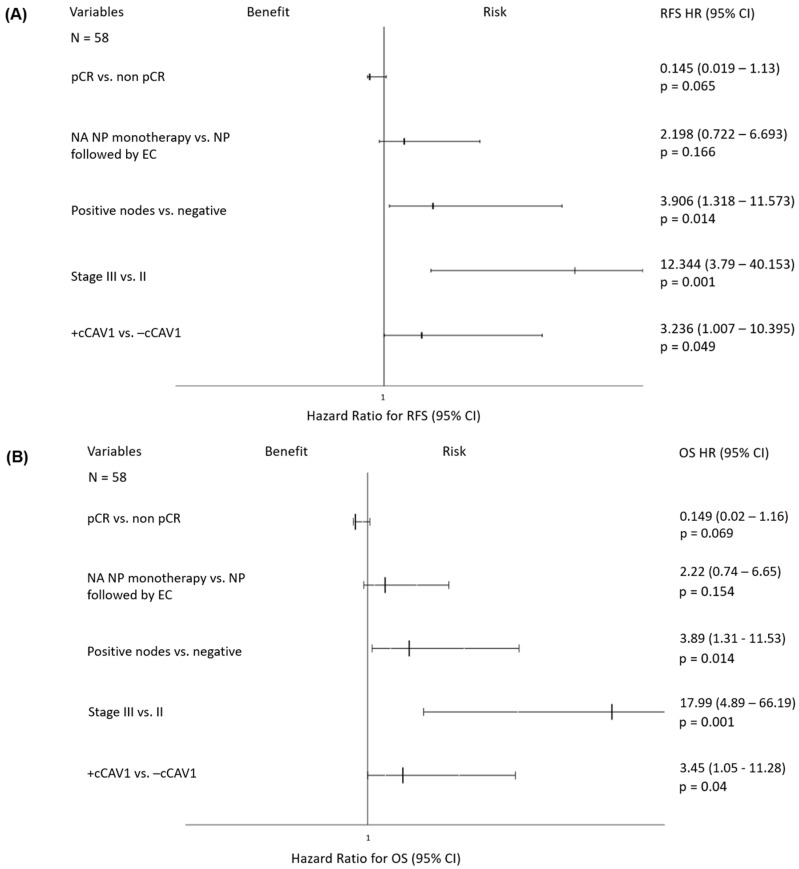
Forest plot multivariant analysis. (**A**) RFS, (**B**) OS. Hazard ratios (HR) and *p*-values were calculated using the log-rank test, with significance set at 0.05. Abbreviations: pCR—pathological complete response; NA—neoadjuvant; NP—nab-paclitaxel; EC—epirubicin and cyclophosphamide.

**Figure 3 ijms-25-12241-f003:**
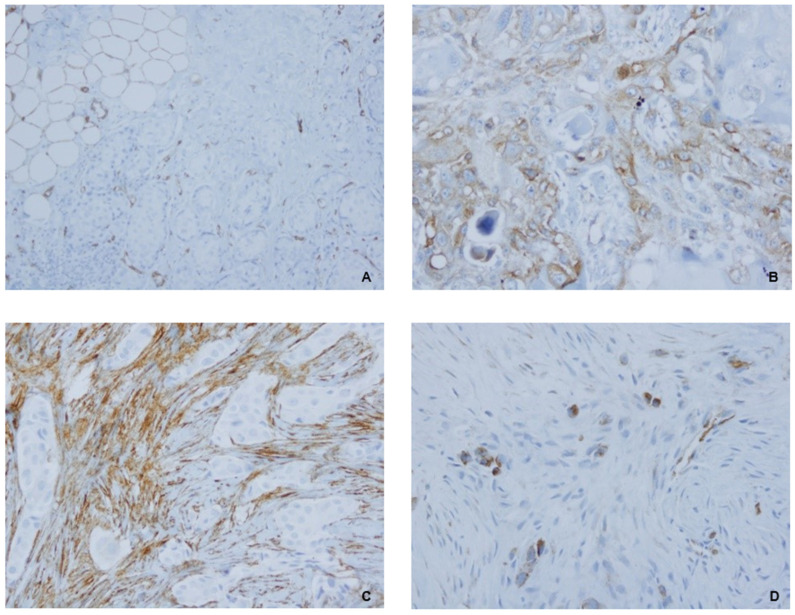
Primary TNBC samples, CAV1 IHC (Caveolin-1 (D46G3) XP Rabbit mAb). (**A**) Positive control ×400; (**B**) positive cCAV1 expression ×400; (**C**) positive sCAV1 expression ×400; (**D**) positive cCAV1 expression ×400.

**Table 1 ijms-25-12241-t001:** Cohort characteristics.

Variable	Entire Cohort (N = 58)
**Age** (y), Mean (min–max)	52.4 (25–83)
**Menopausal status, n (%)** Yes No Unknown	30 (51.7) 27 (46.6) 1 (1.7)
**Clinical Stage, n (%)** II III	42 (72.4) 16 (27.6)
**Nodal status, n (%)** Negative Positive	23 (39.7) 35 (60.3)
**Ki67 mean (min-max)**	68.7 (20–95)
**Histological grade, n (%)** II III Unknown	9 (15.5) 48 (82.8) 1 (1.7)
**TILS mean (standard deviation)**	24.12 (26.6)
**Chemotherapy, n (%)** NP—EC NP	12 (20.7) 46 (79.3)
**Adjuvant Radiotherapy, n (%)** Yes No	49 (84.5) 9 (15.5)

Abbreviations: TILS—tumor-infiltrating lymphocytes; NP—nab-paclitaxel; EC—epirubicin and cyclophosphamide.

**Table 2 ijms-25-12241-t002:** Univariant analysis, table RFS and OS. Hazard ratios (HR) and *p*-values were calculated using the log-rank test, with significance set at 0.05.

	RFS HR (95%CI)	*p*	OS HR (95% CI)	*p*
**Menopausal status** Premenopausal vs. menopausal	1.02 (0.36–2.81)	0.97	1.28 (0.45–3.6)	0.63
**Clinical Stage** III vs. II	5.61 (1.98–15.89)	**0.001**	18.77 (5.15–68.43)	**0.001**
**Node involvement** Positive vs. Negative	3.51 (1.20–10.30)	**0.022**	3.46 (1.18–10.15)	**0.023**
**Histological grade** Grade 2 vs. 3	1.39 (0.31–6.19)	0.66	1.27 (0.28–5.64)	0.75
**Chemotherapy** NP vs. NP-EC	2.99 (1.06–8.42)	**0.038**	2.93 (1.04–8.27)	**0.041**
**Pathological Response** pCR vs. non-pCR	0.116 (0.01–0.88)	**0.038**	0.119 (0.01–0.90)	**0.04**
**Adjuvant Radiotherapy** Yes vs. No	0.60 (0.17–2.14)	0.43	0.60 (0.17–2.15)	0.43
**cCAV1** +cCAV1 vs. −cCAV1	4.96 (1.56–15.71)	**0.006**	5.21 (1.61–16.81)	**0.006**
**sCAV1** +sCAV1 vs. −sCAV1	0.64 (0.22–1.9)	0.43	0.7 (0.23–2.05)	0.516

Abbreviations: RFS—relapse-free survival; OS—overall survival; NP—nab-paclitaxel; pCR—pathological complete response; EC—epirubicin and cyclophosphamide.

## Data Availability

The raw data supporting the conclusions of this article will be made available by the authors on request.

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
