# Peer review of "Assessing the Prognostic Value of Cytoplasmic and Stromal Caveolin-1 in Early Triple-Negative Breast Cancer Undergoing Neoadjuvant Chemotherapy"

_ijms, 2024, doi:10.3390/ijms252212241_

Round 1

Reviewer 1 Report

Comments and Suggestions for Authors

The overall approach is clear, and the results are definite. A few awkward sentences and typos need further correction. The small sample size may limit the applicability of these results to a larger population. The follow-up period is less than five years, which will affect the final results of the survival analysis.

Authors need to clarify the following questions:

1.  Why do patients not meet the 5-year (60 months) follow-up in survival analysis? Currently, breast cancer survival analysis requires at least five years of follow-up.   2. The samples from the patients all come from slices after NAC, making it impossible to determine whether the changes in CAV1 are due to the effects of chemotherapy on the microenvironment or other reasons. At least they should come from tissues obtained before treatment biopsies.   3. The discovery of CAV1 is intriguing; however, the author mentions BRCA1 mutations, and the final analysis shows that CAV1 is unrelated to BRCA1. At this point, what role CAV1 plays in breast cancer, thus becoming a potential biomarker, remains an ambiguous question. Comments on the Quality of English Language

OK

Author Response

Comment 1

The overall approach is clear, and the results are definite. A few awkward 
sentences and typos need further correction. The small sample size may limit 
the applicability of these results to a larger population. The follow-up period is 
less than five years, which will affect the results of the survival analysis.

Reply:
Thank you for your revision. We have carefully revised the typus and grammatical 
coherence of our manuscript. We agree that the small sample size may limit the 
impact of our research results, but we are recruiting more patients that will be 
analysed in a follow-up work. In addition, we are also extending our follow-up of the 
cohort. These limitations have been now mentioned in the discussion of the 
manuscript (line 318-320).

Authors need to clarify the following questions:

Question 1
1. Why do patients not meet the 5-year (60 months) follow-up in survival 
analysis? Currently, breast cancer survival analysis requires at least five years 
of follow-up. 

Reply 1:
We agree that in BC survival analysis used to be 60 months, but we want to clarify 
that our cohort is all with triple negative patient. In early stage of TNBC trials the 
median FUP is generally around 3-5 years, as in the present study: Schmid, P., et al, 
Pembrolizumab for early triple-negative breast cancer. New England Journal of 
Medicine, 382(9), 810–821. https://doi.org/10.1056/NEJMoa1910549

Indeed, median RFS and OS of our cohort are 60.22 (95%CI 54.1 – 66.34) and 62.01 
months (95%CI 56.57 – 67.44) respectively (line 192). However, four out of +cCAV1 
cases relapsed during the follow up period, resulting in a mediant follow-up of 51.43 
months. We have clarified that issue in the results section (line 191-192).

Question 2:
2. The samples from the patients all come from slices after NAC, making it 
impossible to determine whether the changes in CAV1 are due to the effects of 
chemotherapy on the microenvironment or other reasons. At least they should 
come from tissues obtained before treatment biopsies. 

Reply 2:
We want to clarify that all samples are treatment naïve. Sorry for the 
misunderstanding, we have modified the description (line 115).

Question 3:
3. The discovery of CAV1 is intriguing; however, the author mentions BRCA1 
mutations, and the final analysis shows that CAV1 is unrelated to BRCA1. At this 
point, what role CAV1 plays in breast cancer, thus becoming a potential 
biomarker, remains an ambiguous question.

Reply 3:
We agreed with the reviewer, our results do not have sufficient statistical power to 
completely unravel the CAV1role as a biomarker in triple negative breast cancer. 
However, as it has been widely described, only a little percentage of TNBC harbours 
BRCA1/2 mutations. The BRCA mutated cases in our study are very limited and it 
difficult to assess the association between this mutation and CAV1 expression in 
our study. However, and beyond BRCA1/2 mutations, it has been described that 
CAV1 plays a critical role in the progression of breast cancer including cell 
proliferation, apoptosis, autophagy, invasion or migration." (Caveolin-1: a 
multifaceted driver of breast cancer progression and its application in clinical 
treatment. PMID: 30881011).
Briefly,
• CAV1 can affect breast cancer cell proliferation rate by 1) influencing the 
expression and activation of ion channels and receptors on the cell 
membrane 2) regulating cell cycle arrest and affecting mitosis process and 
3) mediating the interaction between the TME (hypoxia conditions and CAFs) 
and breast cancer cells.
• CAV1 can regulate intrinsic and extrinsic apoptotic pathway proteins, such 
as Bcl-2, Bax and caspase-3; 2) CAV1 induced autophagy alternation that 
also can influence cell apoptosis.
• CAV1 is implicated in various aspects of epithelial to mesenchymal 
transition that plays a critical role in breast cancer progression and 
metastasis.
We have included these issues in the discussion section (line 296 – 319). 

Reviewer 2 Report

Comments and Suggestions for Authors

The authors present a very interesting work.

TNBC is demanding targeted approaches and these studies are always welcome and promising, stimulating readers and arousing interest.

Caveolin has been described recently, and several types of cancer are being tested. The study has a solid introduction, and the rationale is logical.

M&M are well described allowing for its replication. Histological pictures are of good quality. Tables are good

References are updated.

Discussion is fair and complete

My decision is to accept

Author Response

Reviewer's comments:

The authors present a very interesting work.
TNBC is demanding targeted approaches and these studies are always
welcome and promising, stimulating readers and arousing interest.
Caveolin has been described recently, and several types of cancer are being 
tested. The study has a solid introduction, and the rationale is logical.
M&M are well described allowing for its replication. Histological pictures are of 
good quality. Tables are good
References are updated.
Discussion is fair and complete
My decision is to accept 

Reply:

Thanks for the comments, we really appreciate them.

Reviewer 3 Report

Comments and Suggestions for Authors

The authors analyzed the expression of CAV-1 in samples from triple negative breast cancer TNBC. This subtype of breast cancer is diagnosed in 10-15% of breast cancer cases, and it is the deadliest type of breast cancer with no targeted treatment available. They chose Caveolin-1 (CAV1) which has been implicated in breast cancer oncogenesis and metastasis and may be a potential prognosticator, especially for non-distant events. CAV1 functions as a master regulator of membrane transport and cell signaling. Several CAV1 SNPs have been linked to multiple cancers, but the prognostic impact of CAV1 SNPs in breast cancer remains unclear.

They showed that the patients who had expression of CAV – 1 in tumor had shorter relapse free survival and shorter overall survival. Expression of CAV – 1 in stroma resulted in enrichment of tumors with lower levels of Ki67and less TILs. 

This is interesting study on TNBC showing the possible mechanisms of TNBC carcinogenesis. Such studies try to find biomarkers of disease progression and targets for better therapeutic strategy.

Author Response

Reviewer's comments:

The authors analyzed the expression of CAV-1 in samples from triple negative 
breast cancer TNBC. This subtype of breast cancer is diagnosed in 10-15% of 
breast cancer cases, and it is the deadliest type of breast cancer with no 
targeted treatment available. They chose Caveolin-1 (CAV1) which has been 
implicated in breast cancer oncogenesis and metastasis and may be a 
potential prognosticator, especially for non-distant events. CAV1 functions as 
a master regulator of membrane transport and cell signaling. Several CAV1 
SNPs have been linked to multiple cancers, but the prognostic impact of CAV1 
SNPs in breast cancer remains unclear.
They showed that the patients who had expression of CAV – 1 in tumor had 
shorter relapse free survival and shorter overall survival. Expression of CAV – 1 
in stroma resulted in enrichment of tumors with lower levels of Ki67and less 
TILs.
This is interesting study on TNBC showing the possible mechanisms of TNBC 
carcinogenesis. Such studies try to find biomarkers of disease progression and 
targets for better therapeutic strategy.

Reply: 

Thanks for your comments, we appreciate them all!